# Multicomponent mapping of boron chemotypes furnishes selective enzyme inhibitors

Joanne Tan[1], Armand B. Cognetta III[2], Diego B. Diaz [1], Kenneth M. Lum[2], Shinya Adachi[1], Soumajit Kundu[2], Benjamin F. Cravatt [2] & Andrei K. Yudin[1]

Heteroatom-rich organoboron compounds have attracted attention as modulators of enzyme function. Driven by the unmet need to develop chemoselective access to boron chemotypes, we report herein the synthesis of α- and β-aminocyano(MIDA)boronates from borylated carbonyl compounds. Activity-based protein profiling of the resulting β-aminoboronic acids furnishes selective and cell-active inhibitors of the (ox)lipid-metabolizing enzyme α/β-hydrolase domain 3 (ABHD3). The most potent compound displays nanomolar in vitro and in situ $IC_{50}$ values and fully inhibits ABHD3 activity in human cells with no detectable cross-reactivity against other serine hydrolases. These findings demonstrate that synthetic methods that enhance the heteroatom diversity of boron-containing molecules within a limited set of scaffolds accelerate the discovery of chemical probes of human enzymes.

[1] Davenport Research Laboratories, Department of Chemistry, University of Toronto, 80 St George Street, Toronto, ON, Canada M5S 3H6. [2] Department of Molecular Medicine, The Skaggs Institute for Chemical Biology, The Scripps Research Institute, 10550N. Torrey Pines Rd., La Jolla, CA 92037, USA. Joanne Tan and Armand B. Cognetta III contributed equally to this work. Correspondence and requests for materials should be addressed to A.K.Y. (email: ayudin@chem.utoronto.ca)

Boron-containing molecules (BCMs) have found widespread utility as mechanistic probes and therapeutic agents[1–6]. The breakthrough discoveries of proteasome inhibitors bortezomib[7], ixazomib[8], and delanzomib[9] opened doors for synthetic and medicinal chemists to explore the potential of boron in therapeutic intervention. Akin to all molecules designed to interact with protein targets, the structures of bioactive BCMs must contain a sizable proportion of heteroatoms, including nitrogen, oxygen, halogens, and sulfur. In the past, the absence of heteroatoms in BCMs has resulted in lack of selectivity amongst related families of enzymes[10]. BCMs display a reversible-covalent mode of inhibition with serine proteases[5]. Boron has a unique ability to adopt a range of coordination modes upon interaction with protein targets[11]. This stands in contrast to other electrophiles such as epoxides, aziridines, and Michael acceptors, which display a singular type of interaction with active site nucleophiles[12–14]. Despite the versatility and recent successes of BCM-driven medicinal chemistry, there are still few examples of boron-containing therapeutic agents. This can be partially explained by the fact that synthetic technologies to site-selectively introduce boron into heteroatom-rich environments remain underdeveloped. The thermodynamic preference of boron to migrate from carbon to oxygen or nitrogen, further aggravated by the low kinetic barrier for these transformations[15,16], accounts for the dearth of available methods.

The structures of the most celebrated boron-containing chemotherapeutics currently on the market are based on the α-aminoboronic acid motif. Inspired by the impact of β-amino acids and β-peptides on contemporary science[17,18], we questioned the potential significance of homology in aminoboronic scaffolds and turned to amphoteric boron-containing compounds as the enabling building blocks. The goal of our work was to build upon facile α-aminoboronic acid synthesis[19,20] and the recently demonstrated stability of β-aminoboronic acids[21].

Herein, we set out to prepare molecular frameworks containing boron and heteroatoms of biological significance in order to map the vicinity of the electrophilic boron warhead. For the design of small heteroatom-rich BCMs, we wanted to ensure that the parent scaffolds featuring mainly hydrogens off the connecting chain could be perturbed by the smallest possible carbon substituent capable of productive interactions with proteins. The chemically robust nitrile functionality came to our attention. Nitriles are not readily metabolized[22–26] and feature a short triple bond. The rod-like nitrile geometry provides a carbon-based substituent with a minuscule steric demand: based on A-values, the CN unit is eight times smaller than the methyl group[27]. This enables nitriles to project into narrow clefts in proteins and engage in productive polar interactions and/or hydrogen bonds in sterically challenging environments[28]. To append nitrile groups to the chain connecting boron and nitrogen, we employ borylated iminium ions of the recently developed α-boryl aldehydes[29–31] and acylboronates[32]. The synthetic utility of borylated iminium ions is derived from the N-methyliminodiacetic acid (MIDA) substituent on boron, which mitigates boron's propensity to undergo C–to–O and C–to–N migrations and enables cellular permeability. Empowered with this strategy, we have identified selective and cell-active inhibitors of the (ox)lipid-metabolizing enzyme α/β-hydrolase domain 3 (ABHD3).

## Results

### Synthesis of the MIDA boronate library.
We began our investigation with a model study shown in Fig. 1. Our recent studies of borylated iminium ions[21] opens doors to run a wide range of multicomponent reactions with potential to expand the accessibility of heteroatom-rich organoboron compounds. Condensation

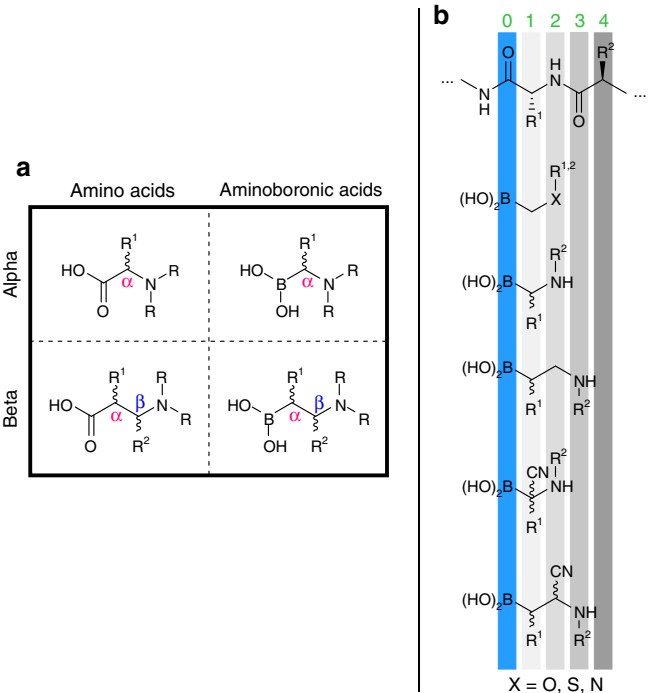

**Fig. 1** An overview of the α- and β-aminoboronic acids investigated in this study. **a** A comparison of the structural similarities between amino acids and aminoboronic acids. **b** A comparison of the structural relations between the peptide backbone and the aminoboronic acid frameworks investigated in this work

of α-boryl aldehydes **1a–g** with primary amines led to borylated iminium ions **2a–g** (Fig. 2a). Under reductive conditions, we were able to access β-amino(MIDA)boronates **3** containing the BCH$_2$CH$_2$N connectivity (Supplementary Methods)[21], which have been shown to play important roles in catalysis[33]. Building on this observation, we were curious to evaluate the behavior of borylated iminium ions in multicomponent reactions. The cyanide anion was chosen as the nucleophilic carbon-based component due to its minuscule steric demand. Upon addition to borylated iminium **2a**, we were excited to see evidence of β-aminocyano(MIDA)boronate **4** formation containing the BCH$_2$CH(CN)N motif (Fig. 2b). During the reaction optimization, we initially hypothesized that benzoyl and acetyl cyanide could act as nucleophilic sources of cyanide[34]. However, the hydrolysis of the iminium ion was faster than the cyanide addition. We also tried using acetone cyanohydrin with catalytic triethylamine[35] and observed desired compound **4** as the major product, but with poor conversion. We then identified trimethylsilyl cyanide as the most suitable reagent when used in the presence of a Lewis acid[35]. Lowering the reaction temperature resulted in poor solubility of α-boryl aldehyde **1a**. The use of 3Å molecular sieves to remove water had no influence on the reaction. The reaction proceeded similarly in both acetonitrile and dichloromethane. We chose to continue our studies with acetonitrile because it improved the solubility of α-boryl aldehyde **1a**.

To be useful as chemical probes, β-aminocyano(MIDA) boronates need to be stable in the free boronic acid form. We anticipated that the nucleophilicity of the secondary nitrogen's lone pair might affect this stability. Indeed, this was the case when we attempted to deprotect MIDA under various mildly acidic and basic conditions. Therefore, we decided to attenuate the nucleophilicity of the nitrogen's lone pair by trapping it as an amide. The β-aminocyano(MIDA)boronates were successfully acylated using acid chlorides and highly electrophilic anhydrides

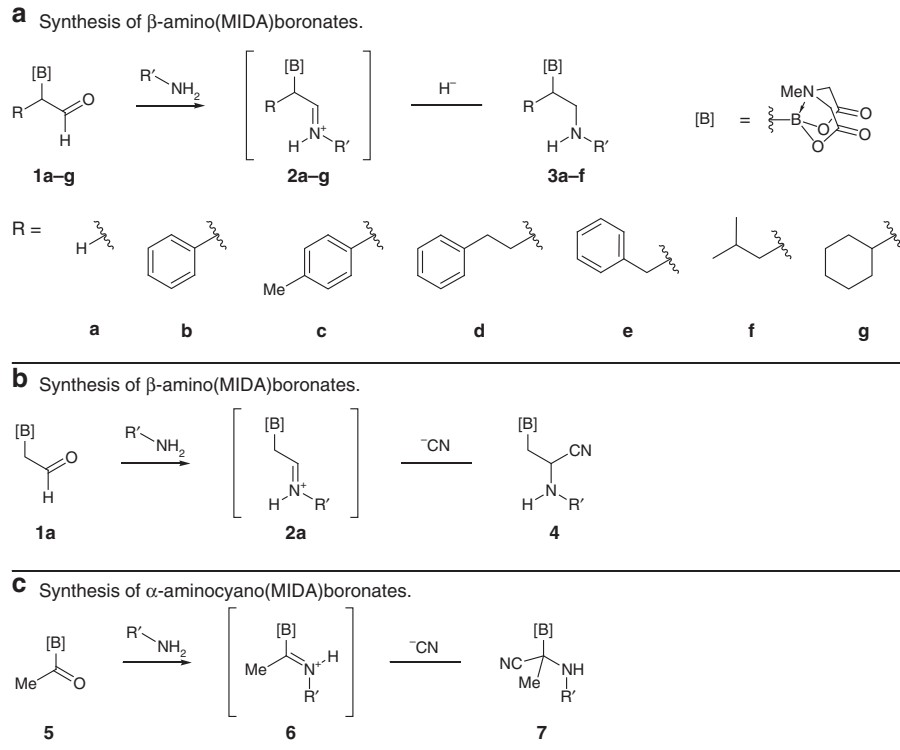

**Fig. 2** General synthesis of the MIDA boronates developed in this study. **a** Synthesis of β-amino(MIDA)boronates using a reductive amination approach. **b** Synthesis of β-aminocyano(MIDA)boronates using a reductive cyanation approach. **c** Synthesis of α-aminocyano(MIDA)boronates using a reductive cyanation approach

such as trifluoroacetic anhydride. Compound **4b** did not react due to its poor nucleophilicity. We were also able to synthesize the β-aminocyano(MIDA)boronates and acylate them in one pot to provide β-aminocyano(MIDA)boronates **4d–l** in higher yields compared to the two-step procedure (81% vs. 55%, respectively for compound **4d**). We also extended this chemistry to acylboronate **5**, to provide α-aminocyano(MIDA)boronate **7**, the B,N-framework that is one carbon chain shorter (Fig. 2c). To further expand the heteroatom mapping set, we prepared the BCO, BCS and BCN motifs (compounds **8–10**) through α-functionalization of alkyl(MIDA)boronates (Supplementary Methods)[20]. A summary of each reaction scope is shown in Fig. 3.

**MIDA boronate screening for serine hydrolase inhibition**. With the β-amino(MIDA)boronates and α- and β-aminocyano (MIDA)boronates in hand, we proceeded to test them for inhibitory activity against serine hydrolases (SHs). SHs represent a large (> 200 member) enzyme family in mammals that plays important roles in metabolizing bioactive proteins, peptides, and metabolites[36]. BCMs are able to interact with the conserved serine nucleophile of SHs through covalent reversible binding, rendering the enzyme inactive[5]. We assessed boronate compounds against a broad set of mammalian SHs by competitive activity-based protein profiling (ABPP) in mouse brain proteome[37]. All library members were initially tested at 20 μM (30 min pre-incubation), and SH inhibition evaluated using the broad-spectrum activity-based probe fluorophosphonate-rhodamine (FP-Rh), where reductions in FP-Rh labeling of one or more SHs in the brain proteome was detected by sodium dodecyl sulfate polyacrylamide gel electrophoresis (SDS-PAGE) and in-gel fluorescence scanning. Multiple members of the boronate library were found to inhibit a low-abundance, ~40 kDa SH previously identified in previous ABPP studies as ABHD3[38] (Fig. 4, Supplementary Fig. 1). Follow-up ABPP of hit compounds screened in a concentration-

dependent manner against HEK 293T lysates recombinantly expressing ABHD3 confirmed boronate inhibition of ABHD3 and identified **4j** as the most potent and selective ABHD3 inhibitor (Supplementary Fig. 2), displaying an in vitro $IC_{50}$ value of 0.14 μM [95% confidence interval (CI, 0.097–0.20 μM)] (Fig. 5a, c). Compound **4j** also inhibited ABHD3 in situ with a similar $IC_{50}$ value [0.040 μM (CI, 0.026–0.060 μM)] (Fig. 5b, c). We also examined the relationship between FP-Rh incubation time and serine hydrolase engagement for **4j** and related compounds, and found that the inhibition profiles were unchanged even at very short time points (5 min), indicating that we did not overlook transient reversible inhibition events in our original 30 min assay (Supplementary Fig. 3). Compound **4j** was evaluated as the racemate. It is likely that further improvements in potency will be uncovered upon evaluation of the enantiomerically pure variants of **4j**. Studies aimed at the synthesis of enantiomerically pure β-aminoboronic acids are currently underway.

**Structure-activity relationship analysis of ABHD3 hits**. All five of the ABHD3 hits contain a phenyl amide, which denotes its importance for the potency. Replacing the phenyl amide with a methyl amide in compounds **4d–f** or the trifluoromethyl amide in **4l** resulted in no notable ABHD3 inhibition. Compound **4j** appears equipotent with compounds **4h** and **4g** against ABHD3, but is more selective against ABHD10. This increased selectivity observed for **4j** may be a result of the highly electronegative fluorine atom, which is known to have profound pharmacological effects when installed in key positions[39].

**Boronate 4j is a selective ABHD3 inhibitor**. ABHD3 is an integral membrane enzyme that is enriched in brain tissue, where it hydrolyzes medium chain and oxidatively modified phospholipids[40]. Few inhibitors have been reported for ABHD3 and these compounds lack selectivity across the SH class[38]. Our initial gel-

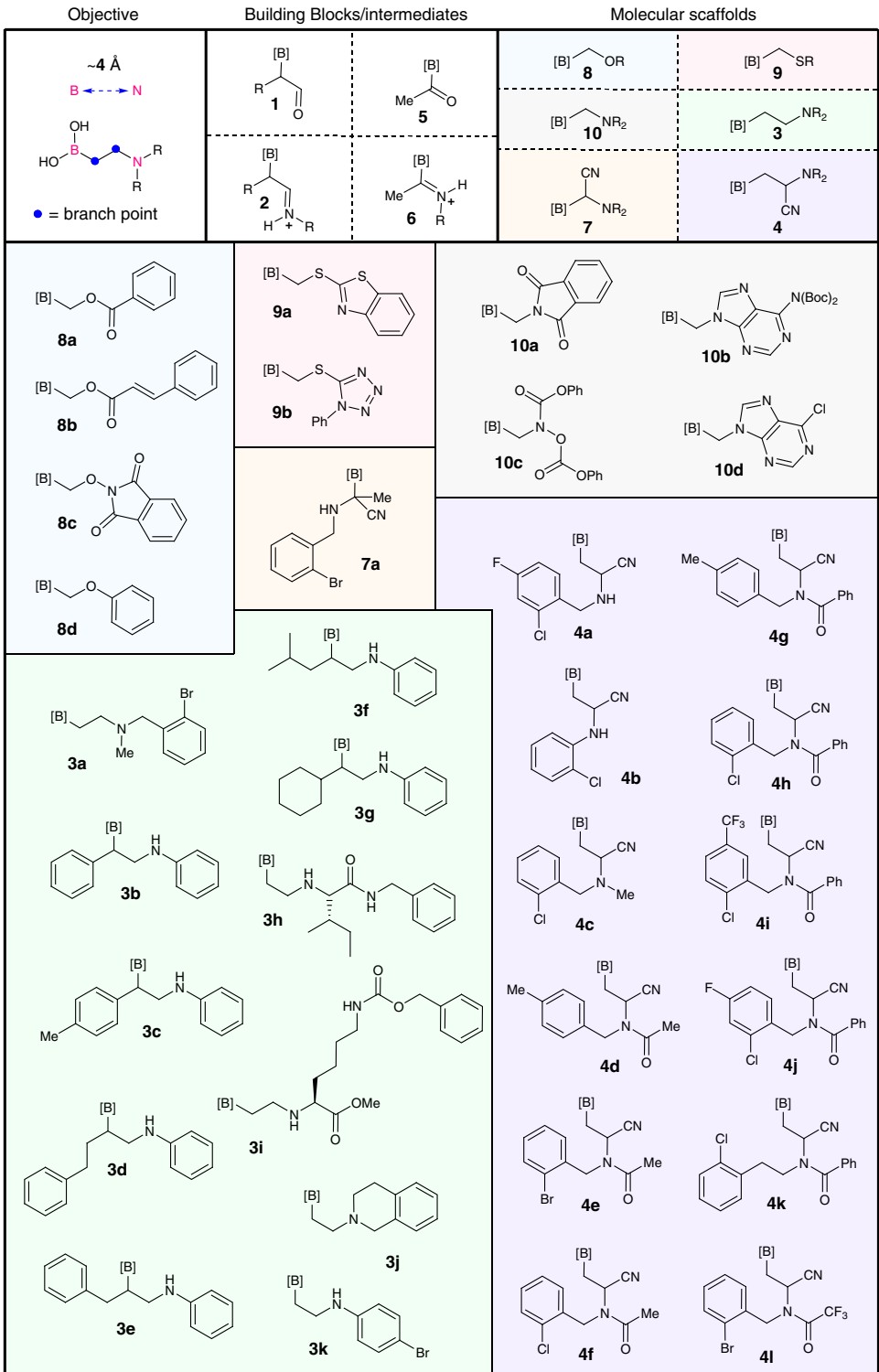

**Fig. 3** MIDA boronate library synthesized and screened for biological activity. An overview of our overall goal, building blocks used, molecular scaffolds we have developed and the respective compounds synthesized in this study. Each panel represents a different chemical reaction and a specific chemotype. The molecules within each panel expand on that chemotype

based ABPP experiments suggested that **4j** exhibited good selectivity for ABHD3 (Fig. 4, Supplementary Fig. 2), but only a limited number of SHs (~20) can be resolved by SDS-PAGE analysis of tissue proteomes. We therefore next evaluated **4j** using higher-resolution, mass spectrometry (MS)-based ABPP methods[41]. We tested **4j** at two concentrations –0.5 and 10 μM – in the

human colon cancer cell line SW620 prepared for quantitative MS-based ABPP using stable isotope labeling with amino acids in cell culture (SILAC)[42]. At both tested concentrations, **4j** fully blocked (95%+) ABHD3 activity without exhibiting any cross reactivity with over 60 additional SHs quantified in SW620 cells (Fig. 5d). Notably, **4j** maintained high specificity for ABHD3 over

all the closely related enzymes and enzyme isoforms within the SW620 cell line; ABHD2, ABHD4, ABHD6, ABHD10–13, ABHD12B, and ABHD16A. These data demonstrate that **4j** is a potent and selective ABHD3 inhibitor that meets the criterion for in situ activity (sub-μM) put forth by the Structural Genomics Consortium[43]. Compound **4j** demonstrates that enhancement of heteroatom diversity of BCMs in a limited set of scaffolds can deliver potent and selective chemical probes. It is tempting to suggest that further expansion of the aminoboronic series to the corresponding γ-aminoboronic acids might lead to additional surprises, allowing one to target other serine hydrolases.

**Investigating mechanism of boronate inhibition of ABHD3.** We further characterized boronate inhibition of ABHD3 by synthesizing a set of **4j** analogues, where the presumed active/ important motifs were removed (Fig. 6a). We prepared an analogue lacking the nitrile group – **11a** – using our previously reported reductive amination protocol[21] (Supplementary Methods). We also synthesized the control without boron **11b** using the same method in Fig. 2b. From compound **4j**, we were able to access the boronic acid analogue **11c** in acidic (10 equiv. HCl in MeOH:MeCN [1:1]) and basic (3 equiv. KOH in MeCN:$H_2O$ [1:1]) conditions. MeCN was necessary to improve solubility. Although the basic conditions fully hydrolyzed **4j** within 2 h, there was also evidence of boric acid, which suggested that **4j** was undergoing protodeborylation. Under acidic conditions, the hydrolysis proceeded cleanly and went to completion after 24 h.

Compound **11a** showed substantially lower potency for ABHD3 (Fig. 6b), underscoring the importance of the cyano group for activity. In contrast, **11c**, which lacked the MIDA protecting group, maintained good ABHD3 inhibition (Fig. 6b), while replacement of the entire MIDA-boronate warhead with an isopropyl group (**11b**) completely ablated ABHD3 inhibition (Fig. 6b). These results support a function for the boron group as an electrophile required for covalent inhibition of ABHD3. However, whether removal of the MIDA protecting group, which might occur in lysates or cells, is required for inhibition of ABHD3 is less clear. The MIDA compound **4j** is slightly more potent in vitro compared to the boronic acid counterpart **11c** (Fig. 6c), and this difference is further magnified in situ, where **4j** exhibited at least 10-fold superior potency (Fig. 6c). These data could indicate that the intact MIDA boronate **4j** is active as an inhibitor of ABHD3, where the boost in cellular activity may reflect enhanced cell permeability or stability of the MIDA-protected over free boronate. Additionally, in contrast with previously described MIDA-boronate inhibitors[20], the MIDA-boronate group of **4j** is largely resistant to hydrolytic cleavage under neutral conditions to the boronic acid (phosphate buffer (pH = 7.4)) in the time frame of the ABPP experiments (Supplementary Fig. 4).

**Metabolomic profiling of 4j.** The discovery that **4j** is a potent, selective, and cell-active inhibitor of ABHD3 prompted us to investigate the effects of pharmacological inhibition of ABHD3 in cells. ABHD3 has been shown to hydrolyze medium chain and oxidatively truncated phosphatidylcholines (PCs)[40], but whether pharmacological inhibition of ABHD3 alters these (or other) lipid metabolites in human cells remains unknown. We performed untargeted metabolomics analysis[44] of SW620 cells treated with **4j** (2.5 μM, 4 h) using a quadrupole time-of-flight (Q-TOF) mass spectrometer. Cells treated with **4j** showed elevations in a specific set of metabolites with $m/z$ values predicted to match medium chain PCs (Fig. 7a). We confirmed these changes for a representative PC species (26:0) by targeted analysis using a triple quadrupole (QQQ) mass spectrometer (positive ion mode,

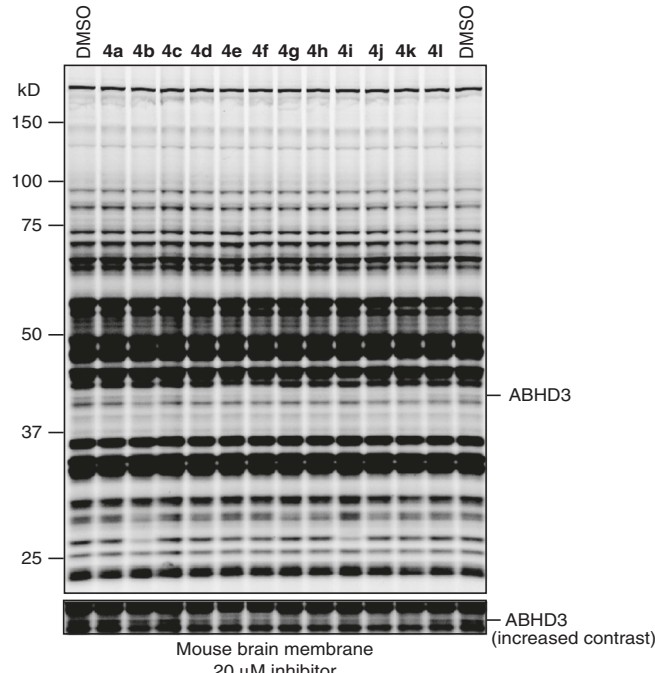

**Fig. 4** MIDA boronates inhibit ABHD3 in mouse brain proteome. ABPP gel of the membrane fraction of mouse brain treated with compounds **4a**–**4l** (20 μM, 30 min, 37 °C), followed by the broad-spectrum, serine hydrolase-directed activity-based probe FP-rhodamine (1 μM, 30 min, RT)

monitoring the PC to phosphocholine transition), which measured an ~5-fold increase in 26:0 PC in cells treated with **4j** (Fig. 7b). We further established the identity of the fatty acyl chains of the PC (26:0) species by targeted analysis in negative mode, where we measured four different transitions corresponding to saturated fatty acids with 10, 12, 14, and 16 carbons, indicating that the 26:0 PC (26:0) was a mixture of two independent, co-eluting species – PC (10:0–16:0) and PC (12:0–14:0). The corresponding fold changes of the matched acyl species indicated that both PCs were elevated by ABHD3 inhibition (Fig. 7c). Taken together, these results indicate that acute pharmacological inhibition of ABHD3 elevates medium-chain PCs in human cells, a finding that is consistent with the metabolic functions previously ascribed to ABHD3 by genetic studies in mice[40].

Using amphoteric boryl building blocks, we have found that homologs of well-known α-aminoboronic acids drastically affect inhibitor selectivity and allow for selective interrogation of lipases. With the help of activity-based protein profiling, we have discovered selective and cell-active inhibitors of the (ox)lipid-metabolizing enzyme ABHD3, with boron as the active warhead. More specifically, we have developed a versatile approach to rapidly assemble boron-containing α-aminonitriles, which have led to the medicinally relevant $BCH_2CH(CN)N$, and BCR(CN)N motifs. The departure from commonly used α-aminoboronic acids allowed us to circumvent protease targets. The remarkable selectivity of compound **4j** among the lipases tested suggests that heteroatom-rich MIDA boronates offer a promising source of serine hydrolase inhibitors. Additionally, the disparity in cellular activities between the MIDA and free boronic acid compounds argues that the MIDA boronate class may exhibit improved cell permeability and/or stability. Our work serves to further exemplify that heteroatom-rich BCMs have the potential to be selective chemical probes.

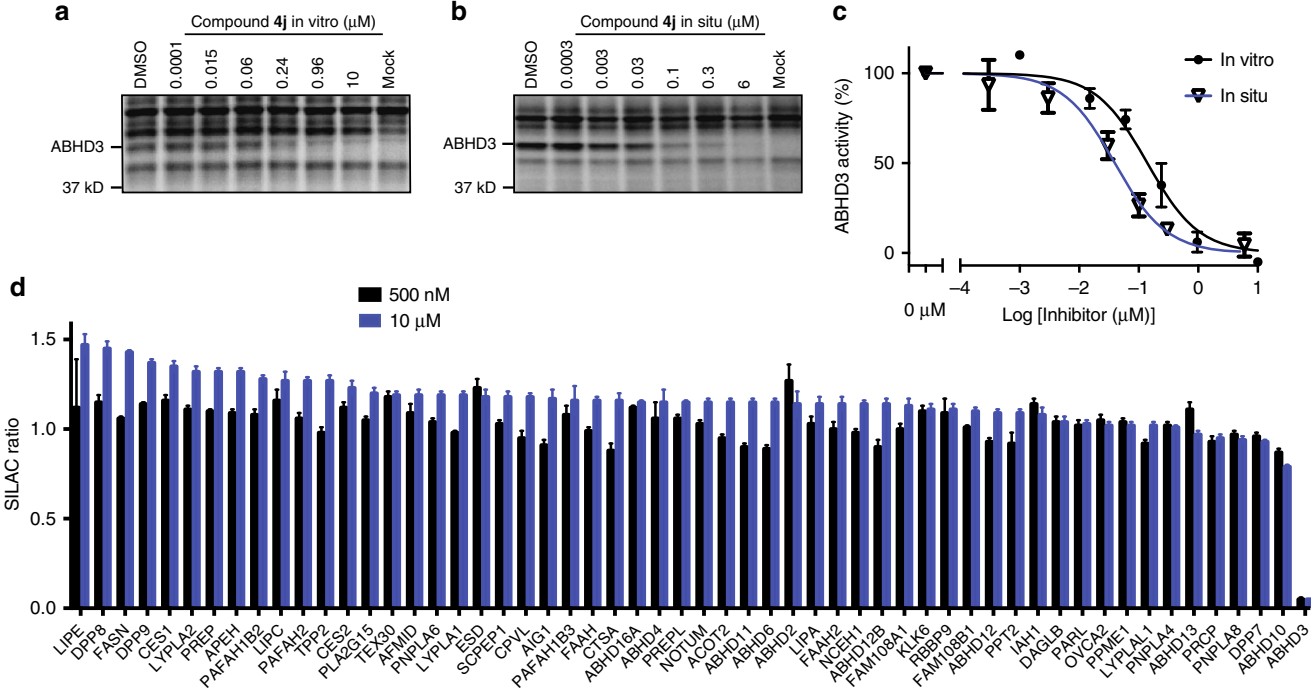

**Fig. 5** Compound **4j** is a potent and selective ABHD3 inhibitor. **a**, **b** ABPP gels of HEK 293T cells overexpressing human ABHD3 treated with compound **4j** (**a**) in vitro (30 min, 37 °C) or (**b**) in situ (2 h, 37 °C). **c** IC$_{50}$ curves for the aforementioned in vitro and in situ **4j** treatments. The data represent average values ± S.E.M. values for three independent experiments per group. **d** Quantitative ABPP-SILAC analysis of SW620 cells treated in situ with DMSO (heavy-isotopically labeled cells) or **4j** (0.5 or 10 μM, 2 h; light-isotopically labeled cells). After in situ treatments, heavy and light cells were lysed, treated with the broad-spectrum, serine hydrolase-directed activity-based probe FP-biotin (6 μM, 1 h, RT), combined and processed for quantitative MS-based analysis, as described[41]. ABHD3 shows complete (95+%) inhibition at both concentrations and **4j** appears to have no major off-targets. The data represent average values ± S.E.M. values for quantified peptides per protein derived from two independent experiments

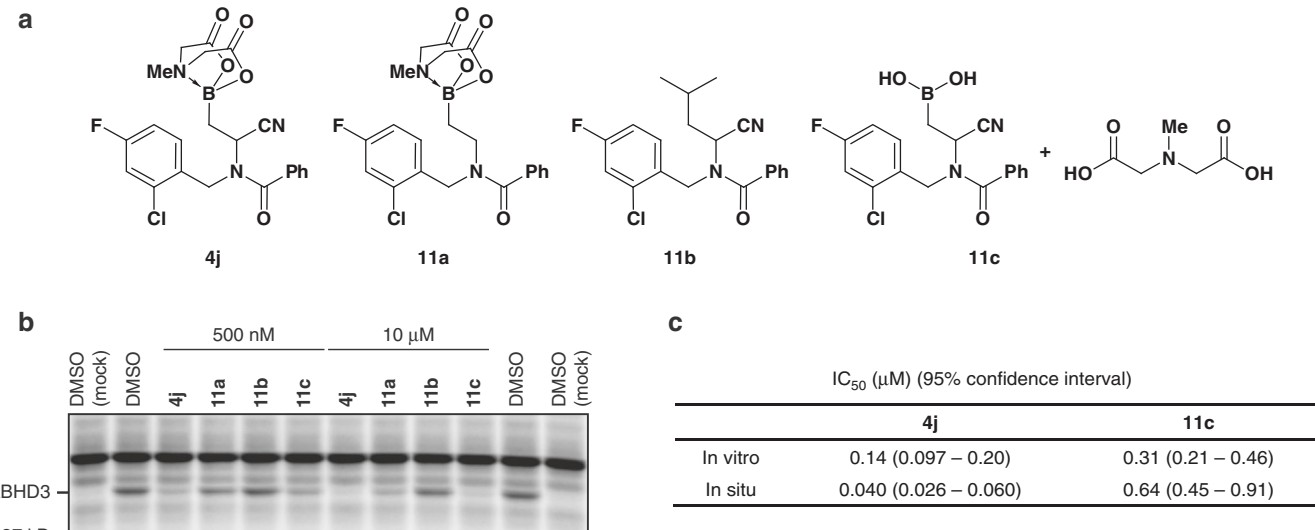

**Fig. 6** Structure-activity relationship for inhibition of ABHD3 by compound **4j**. **a** Structure of **4j** and analogues. **b** ABPP gel of HEK 293T cells overexpressing human ABHD3 treated with **4j** and analogues in vitro. **c** IC$_{50}$ values determined by gel-based ABPP with HEK 293T cells overexpressing human ABHD3. The data represent average values ± 95% confidence intervals from three independent experiments per group

## Methods

**Synthesis of β-aminocyano(MIDA)boronates 4d–4l.** To a flame-dried, round bottom flask equipped with a magnetic stir bar under nitrogen atmosphere was added α-boryl aldehyde **1a** (1.0 equiv.), copper (II) trifluoromethanesulfonic acid (0.05 equiv.) and amine (2 equiv.) in MeCN (0.1 M). After 30 min of stirring,

trimethylsilyl cyanide (2.0 equiv.) was added and then the reaction was left to stir at room temperature until completion. Another equivalent of amine and tri-methylsilyl cyanide was added if the reaction did not go to completion overnight. The reaction was monitored by TLC. Next, catalytic 4-dimethylaminopyridine and N,N-diisopropylethylamine (5.0 equiv.) was added and the reaction was cooled to

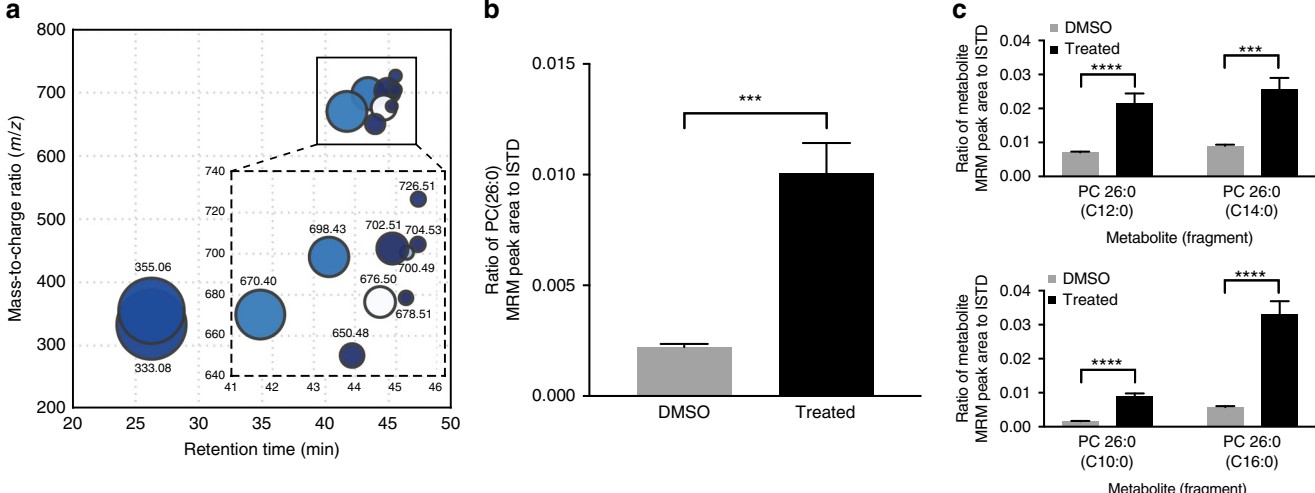

**Fig. 7** Metabolomic profiling of compound **4j** in human cells. **a** SW620 cells were treated with **4j** (2.5 μM, 4 h) or DMSO and collected for metabolomic analysis. Untargeted metabolomics showed a number of significant metabolite changes ($p < 0.01$) measured in positive ionization mode. Bubble sizes represent fold change, while increasing statistical significance is represented by darker shadings. The two bubbles at RT ≈26 min correspond to hydrogen or sodium adduct of a cellular metabolite of **4j** (M = **4j** – $C_5H_6BNO_3$). Several species in the expanded inset were deduced to be potential phosphatidylcholines (PCs) by mass-to-charge ratio and retention time. **b** Multiple reaction monitoring of PC-related transitions (M+H→ phosphocholine) confirmed elevations in 26:0 PC in **4j**-treated cells. **c** Negative mode fragmentation (M+HCO$_2^-$→ RCOO$^-$) revealed PC(26:0) to be a mixture of PC (10:0–16:0) and PC(12:0–14:0) species. For **b** and **c**, the data represent average values ± S.E.M. values from five independent experiments per group. Statistical significance was calculated with unpaired students t-tests comparing **4j**-treated to DMSO-treated groups; ***$p < 0.001$; ****$p < 0.0001$

0 °C. The corresponding acid chloride or anhydride (4.0 equiv.) was added to the solution dropwise or in two separate increments every 2 h. The reaction was stirred at room temperature for 4 h until completion. The solvent was removed by in vacuo and then extracted with ethyl acetate (×3) and brine. The organic layer was collected and then acidified to pH 1 using 0.1 M HCl. The aqueous layer was removed and then saturated NaHCO$_3$ was added to the organic layer until it reached pH 8–9. The organic layer was dried over Na$_2$SO$_4$, filtered, concentrated in vacuo and purified by flash chromatography or CombiFlash using hexanes:acetone to afford pure product.

**Preparation of metabolomics samples.** SW620 cells were seeded at 3 million cells per plate (10 cm) in RPMI media supplemented with Fetal Bovine Serum, penicillin (100 U/mL), streptomycin (100 μg/mL) and L-glutamine (2 mM). 2 days later the cells were washed with PBS (3×) and incubated with serum free RPMI containing either inhibitor (2.5 μM), or DMSO, for 4 h. Cells were then washed with PBS (3×), lysed by adding cold methanol (1 mL) directly to the plate, harvested using a cell scraper, and transferred to a 2 dram vial containing cold CHCl$_3$ (2 mL) and PBS (1 mL). After centrifugation, the bottom, organic layer was collected, and formic acid (50 μL) and additional CHCl$_3$ (2 mL) was added. The solution was vortexed vigorously, followed by centrifugation, and the organic layer was again collected and combined with the first extraction. Solvent was removed under a stream of nitrogen, and samples were stored at −80 °C prior to use.

**Untargeted metabolomics analysis.** Discovery Metabolite Profiling (DMP) was performed as previously described[45]. Briefly, metabolomes were resuspended in 2:1 CHCl$_3$/MeOH and 30 μl was injected onto an Agilent 6520 series quadrupole-time-of-flight (Q-TOF) MS. LC separation was achieved using a Gemini reverse-phase C18 column (5 μm, 4.6 mm × 50 mm, Phenomonex). Mobile Phase A consisted of H$_2$O/MeOH (95:5) and Mobile Phase B of IPA/MeOH/H$_2$O (60:35:5) with additional formic acid (0.1%) or saturated aqueous ammonium hydroxide (0.1%) in positive and negative modes, respectively. The LC method consisted of 0.1 ml/min 0% buffer B for 5 min, a 0.4 ml/min linear gradient over 40 min to 100% buffer B, 0.5 ml/min 100% buffer B for 10 min, and 0.4 ml/min equilibration with 0% buffer B for 5 min, for an overall run time of 60 min. MS analysis was performed with an ESI source in scanning mode from $m/z = 150 − 1200$.

The capillary voltage was set to 4.0 kV and the fragmentor voltage was set to 100 V. The drying gas temperature was 350 °C, the drying gas flow rate was 11 l/min, and the nebulizer pressure was 35 psi. Analysis of the LC-MS data was performed using XCMS (https://xcmsonline.scripps.edu/).

**Targeted MRM measurements of phospholipid and lysophospholipid species.** Metabolomic profiling experiments were performed in positive mode as previously described[40] using an Agilent 6460 series Triple Quad (QQQ) using the same LC separation and buffers as described above. A full list of targeted PCs,

lysophosphatidylcholines, phosphatidylethanolamines, and lysophosphatidylethanolamines, can be found in Long et al.[40]. All metabolites were fragmented with a collision energy of 30 V.

**Data availability.** The data that support the findings of this study are presented within the article and its Supplementary Information file and from the corresponding author upon reasonable request. All constructs originally described in this study can be obtained and used without limitations for non-commercial purposes on request from the corresponding author.

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

## Acknowledgements

We thank the Canadian Foundation for Innovation, project number 19,119, the NIH (DA033760), and the Ontario Research Fund for funding of the Centre for Spectroscopic Investigation of Complex Organic Molecules and Polymers. We also thank Dr. Darcy Burns and Mr. Dmitry Pichugin for their assistance with NMR spectroscopic experiments. We thank the Natural Science and Engineering Research Council (NSERC) and the Canadian Institutes of Health Research (CIHR) for financial support.

## Author contributions

J.T.: Optimized and carried out the modified Strecker experiments, acquired and analyzed the spectroscopic data for the α- and β-aminocyano(MIDA)boronates, boronic acids and non-boron containing analogues and prepared most of the manuscript and Supplementary Information with an equal contribution from A.B.C. III. D.B.D.: Performed the reductive amination experiments, acquired and analyzed the spectroscopic data for the β-amino(MIDA)boronates and helped prepare some of the Supplementary Information. S.A.: Synthesized compounds **8**–**10**. A.B.C. III also performed ABPP, MS, and metabolomics experiments. S.K.: Performed the initial ABPP screening of the MIDA boronate library. K.M.L.: Helped perform and analyze the metabolomic experiments. A.K.Y.: Conceived the initial ideas of this work, helped prepare the manuscript and provided guided the chemistry experiments. B.F.C.: Helped guide proteomic and metabolomic experiments and prepare the manuscript.

## Additional information

**Competing interests:** The authors declare no completing financial interests.

