## [Peer Review File · Nature Communications]

Reviewers' comments:

Reviewer #1 (Remarks to the Author):

The paper describes the generation of a novel 'heteroatom-rich' library of boronate derivatives, and the subsequent use of this library in the discovery of selective inhibitors of serine hydrolase ABHD3 enzyme using a proteome approach followed by more detailed in vitro assays. In addition to the ability of members of this library to selectively inhibit ABHD3, the paper reveals that selected compounds also display marked cellular potency. The authors were also able to make comment as to the SAR within the library in terms of binding to the enzyme, but also in terms of factors that may effect cellular penetration.

The experimental part of the work appears to be complete and to a high standard. One important aspect that was not commented upon by the authors was the fact that, apparently, the majority of the compounds within the library are chiral, although the chemistry described for the synthesis of these will yield racemates and, presumably, all chiral compounds were screened as racemates. It would have been useful for the authors to have commented on this aspect as it is likely that the enantiomeric versions of the inhibitors may show greater efficacy towards the enzyme than that for the racemic forms.

Reviewer #2 (Remarks to the Author):

The manuscript by Yudin and coworkers explores a series of boron chemotypes as serine hydrolase inhibitors. The rationale, design and synthesis is clear, leading the authors to develop a series of stable MIDA boronates for in situ fluorophosphonate profiling. While ABHD3 is quite dim in their gel-based profiling, they later show amazing potency and selectivity in SILAC profiling proteomics studies. Furthermore, they corroborate previous knockout mouse studies showing elevated medium chain PCs. This compound now opens new opportunities to study the role of ABHD3 in cell-based systems. As a minor point, it would be valuable to add a gel showing fluorophosphate labeling at a few different (fluorophosphonate) concentrations or labeling time points. Given the the covalent-reversible nature of inhibition, it is possible that the fluorophosphonate displaces readily reversible inhibitors across different serine hydrolase active sites, potentially missing some weak non covalent interactions. Clearly these compounds are selective and can affect ABHD3 activity in cells. It is not totally clear how challenging it will be to extend this chemotype to other hydrolases, since the only other target of any compound was abhd10. More discussion would be appropriate to project how to expand this approach to hit other enzymes besides abhd10 and abhd3. It would also be useful to see the SILAC MS data, since it is not clear how many peptides were used for quantitation for each data point. Overall, this is a strong manuscript and should be published after minor revisions.

Reviewer #3 (Remarks to the Author):

My review is only based on the synthetic chemistry, which has been diligently carried out, and all of which is based upon now well-established literature precedent.

Referee #1:

1. The paper describes the generation of a novel 'heteroatom-rich' library of boronate derivatives, and the subsequent use of this library in the discovery of selective inhibitors of serine hydrolase ABHD3 enzyme using a proteome approach followed by more detailed in vitro assays. In addition to the ability of members of this library to selectively inhibit ABHD3, the paper reveals that selected compounds also display marked cellular potency. The authors were also able to make comment as to the SAR within the library in terms of binding to the enzyme, but also in terms of factors that may effect cellular penetration. The experimental part of the work appears to be complete and to a high standard.

We are pleased that the Referee is satisfied with the standard we carried out the synthetic experiments.

2. One important aspect that was not commented upon by the authors was the fact that, apparently, the majority of the compounds within the library are chiral, although the chemistry described for the synthesis of these will yield racemates and, presumably, all chiral compounds were screened as racemates. It would have been useful for the authors to have commented on this aspect as it is likely that the enantiomeric versions of the inhibitors may show greater efficacy towards the enzyme than that for the racemic forms.

We agree with the Referee and have included a discussion on how chirality may influence the potency. We are also currently working on developing the enantioselective synthesis of the β -aminocyanoboronates (see highlights #1). This in itself is a significant synthetic undertaking that constitutes a separate and rather intense research project. We are happy to say that we are well on our way to solving this problem and another student from the Yudin lab recently visited Merck Research Laboratories in New Jersey and developed an enantioselective access to derivatives of beta-amino boronic acids. This paper will appear in due course and will serve as an appropriate means to prepare and examine enantiomerically pure versions of probes identified in our present paper.

Referee #2:

1. The manuscript by Yudin and coworkers explores a series of boron chemotypes as serine hydrolase inhibitors. The rationale, design and synthesis is clear, leading the authors to develop a series of stable MIDA boronates for in situ fluorophosphonate profiling. While ABHD3 is quite dim in their gel-based profiling, they later show amazing potency and selectivity in SILAC profiling proteomics studies. Furthermore, they corroborate previous knockout mouse studies showing elevated medium chain PCs. This compound now opens new opportunities to study the role of ABHD3 in cell-based systems. Clearly these compounds are selective and can affect ABHD3 activity in cells. As a minor point, it would be valuable to add a gel showing fluorophosphate labeling at a few different (fluorophosphonate) concentrations or labeling time points. Given the the covalent-reversible nature of inhibition, it is possible that the fluorophosphonate displaces readily reversible inhibitors across different serine hydrolase active sites, potentially missing some weak non covalent interactions.

This is a good suggestion. In the revised manuscript, we show gel-based ABPP data reflecting an analysis of serine hydrolase targets at different time points of FP-Rh treatment (5, 15, and 30 min). This analysis, which is provided in Supplementary Fig. 3, confirmed the boronate compound activity for ABHD3 and ABHD10 at all time points, but did not reveal additional targets even at the shortest FP-Rh incubation times. We interpret these results to indicate that we are not missing weaker reversible inhibition events in our ABPP experiments (see highlights #1).

2. It is not totally clear how challenging it will be to extend this chemotype to other hydrolases, since the only other target of any compound was abhd10. More discussion would be appropriate to project how to expand this approach to hit other enzymes besides abhd10 and abhd3.

We agree with the Referee and believe this is a valid point. We have included a discussion on how this we might expand our approach to other serine hydrolases through the design of γ -aminoboronic acids and chemical manipulation of the nitrile handle (see highlights #2).

3. It would also be useful to see the SILAC MS data, since it is not clear how many peptides were used for quantitation for each data point.

We have added an Excel file showing our full proteomics datasets as well as the filtered SH data, and how it was compiled.

4. Overall, this is a strong manuscript and should be published after minor revisions.

We are glad that the Referee is pleased with our work.

Referee #3:

1. My review is only based on the synthetic chemistry, which has been diligently carried out, and all of which is based upon now well-established literature precedent.

We are happy that the Referee is satisfied with the standard we carried out the synthetic chemistry.

REVIEWERS' COMMENTS:

Reviewer #2 (Remarks to the Author):

The revised manuscript addresses the previous concerns and is ready for acceptance.